# Graves’ Eye Disease: Clinical and Radiological Diagnosis

**DOI:** 10.3390/biomedicines11020312

**Published:** 2023-01-22

**Authors:** Kasen R. Hutchings, Seth J. Fritzhand, Bita Esmaeli, Kirthi Koka, Jiawei Zhao, Salmaan Ahmed, James Matthew Debnam

**Affiliations:** 1Department of Neuroradiology, The University of Texas MD Anderson Cancer Center, Houston, TX 77030, USA; 2Orbital Oncology & Ophthalmic Plastic Surgery, Department of Plastic Surgery, The University of Texas MD Anderson Cancer Center, Houston, TX 77030, USA; 3Orbit, Oculoplasty, Reconstructive and Aesthetic Services, Sankara Nethralaya, Chennai 600006, India

**Keywords:** Graves’ disease, Graves’ eye disease, extraocular muscles, compressive optic neuropathy, lacrimal gland

## Abstract

Graves’ disease is an autoimmune disorder in which hyperthyroidism results in various systematic symptoms, with about 30% of patients presenting with Graves’ eye disease (GED). The majority of patients with GED develop mild symptoms, including eyelid retraction, exposure of the globe, superior rectus–levator muscle complex inflammation, and fat expansion, leading to exophthalmos. More severe cases can result in extraocular muscle enlargement, restricted ocular movement, eyelid and conjunctival edema, and compression of the optic nerve leading to compressive optic neuropathy (CON). GED severity can be classified using the Clinical Activity Score, European Group on Graves’ Orbitopathy scale, NO SPECS Classification system, and VISA system. CT and MRI aid in the diagnosis of GED through the demonstration of orbital pathology. Several recent studies have shown that MRI findings correlate with disease severity and can be used to evaluate CON. Mild cases of GED can be self-limiting, and patients often recover spontaneously within 2–5 years. When medical treatment is required, immunomodulators or radiotherapy can be used to limit immunologic damage. Surgery may be needed to improve patient comfort, preserve the orbit, and prevent vision loss from optic nerve compression or breakdown of the cornea.

## 1. Introduction

Under normal physiologic conditions, thyroid function is regulated by the hypothalamic–pituitary axis. In response to low triiodothyronine (T3) and thyroxine (T4) levels, thyrotropin-releasing hormone (TRH) is released from the hypothalamus, leading to increased stimulation of anterior pituitary gland thyrotrophs and production of thyroid-stimulating hormone (TSH). TSH is secreted into the blood and binds to the thyroid-stimulating hormone receptor (TSHR), causing the release of T3 and T4 from thyroid follicular cells. Through negative feedback, TRH levels decrease. This system allows the body to maintain thyroid hormone homeostasis.

This homeostasis is radically disrupted in Graves’ disease, the most common etiology of hyperthyroidism in the developed world. Graves’ disease is caused by a breakdown of immunological tolerance and the subsequent formation of antibodies that target the TSHR. These antibodies bind to TSHRs and overstimulate the release of T3 and T4 regardless of TRH or TSH levels. The elevation of thyroid hormones may cause thyrotoxicosis and deposition of autoreactive lymphocytes within thyroid tissue [1]. Additionally, the severity of immune tolerance breakdown is enhanced as the function of the TSHR, thyroid peroxidase (TPO), and thyroglobulin (Tg) is altered [2].

Graves’ disease is 5–10 times more common in women and typically presents between 30 and 60 years of age [3]. An estimated 79% of the risk for Graves’ disease can be attributed to a genetic predisposition. The other 21% results from environmental and occupational risk factors such as smoking, radiation exposure, stress, iodine toxicity, deficiency of vitamin D, and Agent Orange exposure [4]. Viruses including Hepatitis C and Epstein–Barr may also put patients at risk for Graves’ disease [3]. 

Many Graves’ disease symptoms are caused by elevated T3 and T4, including fatigue, heat intolerance, palpitations, weight loss, anxiety, sleep disturbance, polydipsia, and sweating [5,6]. Patients can present with a nodular goiter leading to esophageal or tracheal compression and resulting dysphagia, orthopnea, or globus pharyngeus. Graves’ disease is also associated with thyroid dermopathy, a rare disorder characterized by lesions of thickened, pigmentated skin, often involving the pretibial region [7]. There might also be clubbing of the digits and toes due to acropachy, another rare extrathyroidal manifestation [8,9]. 

## 2. Graves’ Eye Disease

Patients with Graves’ disease are also at risk of developing Graves’ eye disease (GED), an inflammatory condition of orbital soft tissues that occurs in approximately 25% of cases [10,11]. Though the exact mechanism of this disease remains unclear, a cross-reaction of TSHR antibodies and antigens appears to cause lymphocyte infiltration and activation within orbital tissue [12]. As GED progresses, inflammatory cells such as lymphocytes, plasma cells, macrophages, and eosinophils infiltrate the extraocular muscles (EOMs). Chronic GED may be further complicated by collagen deposition and fibrosis [13]. Although 90% of patients with GED have hyperthyroidism, a small percentage of patients can have euthyroidism or hypothyroidism [14].

GED is the most prevalent orbital disease, with a rate of 2% among women and 0.5% among men. It has been shown to affect all races [15]. In 70% of patients with GED, the symptoms present within 12 months of Graves’ disease onset. In some cases, GED can be the initial presentation [16,17]. Patients tend to be at a higher risk of developing GED due to smoking, stress, radioactive iodine exposure, or a family history of GED [18]. 

The clinical findings of GED vary based on the location of the affected orbital soft tissue. Most patients develop mild symptoms, which commonly include eyelid retraction, globe exposure, EOM inflammation, and fat expansion, leading to exophthalmos [19]. This clinical GED presentation tends to develop gradually over several months [20]. Approximately 40% of patients with GED present with EOM enlargement. This usually occurs in an older population, leading to the rapid development of more severe symptoms, including restricted ocular movement, optic nerve compression, and edema of the conjunctiva and eyelids. Typically, the more severe symptoms develop in a biphasic pattern beginning with a 6–18 month-long progressive phase and a subsequent inactive phase. Smoking history and family history of GED are often associated with these symptoms [20,21,22].

More than 80% of patients with GED will develop retraction of the upper eyelid [23]. Superior rectus–levator muscle complex enlargement has been shown to correlate with upper eyelid retraction [24]. This connection suggests that this muscle complex is the most commonly affected muscle in GED. However, the inferior rectus muscle has generally been thought to be the most commonly enlarged extraocular muscle in GED [25,26,27]. Research has shown that in GED, involvement of the superior rectus–levator muscle complex is common, although other EOMs were only shown to be involved in approximately one-third of cases. Enlargement of the superior rectus–levator muscle complex in isolation presents in 30% of GED patients [24,28].

Lower eyelid retraction is another symptom of GED and often manifests as exophthalmos and scleral show. Exophthalmos in GED can also occur due to retro-orbital fat expansion. When combined, eyelid retraction and exophthalmos can result in cornea exposure, irritation, and ulceration. These patients will also be at risk for eventual visual loss. Periorbital soft-tissue inflammation often presents as discomfort of the orbit, conjunctival injection, orbital edema, and swelling of the eyelids.

In the initial active inflammatory phase, it is common for patients to experience progressive intermittent restriction of ocular motility. This phase is also associated with conjunctival injection, edema of affected muscles, and pain with eye movement. As the disease progresses, secondary fibrosis can lead to motility restriction and resulting diplopia and strabismus [20].

Approximately 6% of GED cases involve compressive optic neuropathy (CON), which is caused by hypertrophied EOMs compressing the optic nerve at the orbital apex [29]. Clinical manifestations can include decreased visual acuity, decreased visual-evoked potentials, relative afferent pupillary defect, and visual field defects [30,31].

## 3. GED Grading Systems

Four primary schemes are used to classify and grade GED severity: the Clinical Activity Score (CAS), European Group on Graves’ Orbitopathy (EGOGO) scale, NO SPECS Classification system, and VISA system.

The CAS uses a summed binary scale to identify patients with GED who could benefit from immunosuppressive therapy (Table 1). This scale uses a point system with points tallied for seven different signs and symptoms of periorbital soft-tissue inflammation. Additional points are recorded during a follow-up evaluation for progression of disease: decreased visual acuity, decreased ocular motility (≥8°), or increased exophthalmos (≥2 mm). The resulting score is highly predictive of immunosuppressive therapy efficacy. CAS scores ≥4 have a positive predictive value of 80% and a negative predictive value of 64% in predicting corticosteroid efficacy [32].

The EGOGO scale categorizes GED severity within one of three disease states: mild disease, moderate-to-severe disease, or very serious disease. The mild disease classification is used in cases with exophthalmos, mild eyelid retraction, and limited EOM involvement. Treatment for GED categorized as mild disease tends to be relatively conservative. GED cases categorized as moderate-to-severe disease present with restricted eye motility, exophthalmos of more than 25 mm, and a greater degree of inflammatory changes. Treatment for this category is usually medical. The most severe EGOGO categorization is very serious disease, including less common conditions such as corneal ulceration and CON. This categorization often requires surgical treatment [33].

The NO SPECS Classification system assigns each case a Global Severity Score, which grades based on GED signs and symptoms (Table 2). This score emphasizes GED features by order of frequency of presentation; however, this system does not provide managemental guidelines or include clinical activity [34].

The VISA system focuses on GED symptoms and physical exam findings by assessing four independently graded parameters: vision, inflammation, strabismus, and appearance. A graded score out of 10 is assigned based on the results (Table 3). Patients with a score of less than 4 are treated conservatively, but a score greater than 5 may require a more aggressive regimen as this score shows evidence of progressive inflammation [18,35].

## 4. Clinical Diagnosis

GED diagnosis is based on three common symptoms: exophthalmos, eyelid retraction, and restricted orbital motility [36]. These features are especially significant when they are found bilaterally. Increased T3 and T4, decreased TSH, or a family history of thyroid disease can support the diagnosis, though it must be noted that approximately 10% of patients with GED have normal or low thyroid hormone levels at disease onset [14]. Diagnosis of Graves’ disease can be confirmed through the detection of TSHR antibodies. Imaging modalities such as ultrasound, CT, and MRI can aid GED diagnosis [37,38]. 

## 5. Radiological Diagnosis 

Radiological studies of patients with suspected GED help confirm the initial clinical diagnosis as they provide a detailed demonstration of orbital pathology. For example, although many GED cases initially present unilaterally, approximately 90% of cases have bilateral involvement, as seen on imaging [39]. These studies can also aid in the assessment of treatment response or progression of disease. 

### 5.1. Computed Tomography (CT)

CT is the preferred method for evaluating the orbital bony structures, including remodeling and post-operative changes following orbital decompression (Figure 1). CT can also accurately demonstrate eyelid edema, lacrimal gland prolapse, and increased orbital fat that can result in optic nerve stretching [40,41]. CT is faster than MRI but uses low-dose radiation. Lymphocyte accumulation and mucopolysaccharide deposition can present with hypodense areas within the EOMs [39]. The Hounsfield unit (HU) density of EOMs can be used to characterize fatty infiltration in patients with GED. Hounsfield units are a measurement of soft-tissue density, with higher HU values corresponding to tissues of greater density. Water is defined as 0 HU, and values range from −1000 HU for air to 700–2000 HU for bone, and 3000 HU for certain metals. Cohen et al. found significant differences in Hounsfield unit density in cases of GED (−40.4 HU) compared to patients without GED (−34.8 HU) (*p* = 0.048) [42]. 

### 5.2. Magnetic Resonance Imaging (MRI)

MRI is better suited than CT for the evaluation of soft-tissue structures due to its high spatial resolution with varying sequences aiding in the diagnosis of GED. MRI offers a greater range of soft-tissue contrast than CT and is better able to depict features that are obscured by bone and metal. T1-weighted and T2-weighted scans constitute the most common MRI sequences. T1-weighted images provide insight into the anatomy of an image, with contrast proving useful for defining the extent of the disease. Fat is hyperintense and appears bright white on T1-weighted sequences, which enables clear visualization of the disease involvement relative to fat. T2-weighted imaging is employed to define the characteristics of the disease. 

On MRI, enlargement of the EOMs is demonstrated, and fatty infiltration causes the EOMs to appear T1 hyperintense, while edema causes them to appear T2 hyperintense (Figure 2). Additionally, superior ophthalmic vein dilation and EOM enhancement may also be noted. During the acute phase of GED, EOM belly enlargement with tendinous insertion sparing can be identified; however, occasional cases include portions of tendinous thickening. Coronal plane imaging can reveal EOM enlargement and optic nerve compression at the orbital apex (Figure 3). 

### 5.3. MRI Correlation with Disease Severity

MRI findings have been correlated with disease severity. Kvetny et al. [43] found that the thickness of EOMs on T1-weighted sequences was significantly associated with the patients’ CAS and the duration of active disease. Tortora et al. [44] studied the signal intensity of the EOMs on the short tau inversion recovery (STIR) and T1-weighted post-contrast sequences and correlated the signal intensities with the Clinical Activity Score (CAS). They found a significant correlation between the signal intensity values on the STIR and T1 post-contrast sequences with the severity of the CAS scores. The authors concluded that MRI could be used to establish the activity phase of GED more than CAS by itself and that contrast-enhanced sequences may not be needed to correlate MRI with CAS. 

Higashiyama et al. [45] studied the signal intensity ratios (SIRs) of the orbital fat and EOMs on STIR sequences in patients with active and inactive GED and controls. The mean SIR in the active GED group was significantly higher than both the inactive group and controls (*p* < 0.001). The SIR of orbital fat in all patients with GED demonstrated a significant positive correlation with that of the EOMs (*p* < 0.001). A correlation was also found between the severity of GED and the orbital fat SIR. The authors concluded that measuring SIRs in orbital fat may be useful in the evaluation of disease activity. 

Ollitrault et al. [46] demonstrated that Dixon-T2-weighted imaging (Dixon-T2WI) has a higher sensitivity (100% vs. 94.7%) and specificity (71.2% vs. 68.5%) for assessing inflamed EOMs in patients with GED as compared to T1, T2, and fat-suppressed T2-weighted MRI sequences at 3 Tesla. In this study, patients with confirmed GED underwent Dixon-TT2WI and conventional MRI sequences. The sequences were randomly and independently interpreted by two neuroradiologists, blinded to all data. Dixon-T2WI was significantly more likely to detect at least one inflamed EOM when compared to conventional sequences (*p* = 0.02) and significantly less likely to show artifacts when compared to fat-suppressed T2WI (*p* < 0.001). Confidence of the diagnosis was significantly higher with Dixon-T2WI than with the conventional sequences (*p* = 0.003).

Wu et al. [47] studied the microstructural and morphological changes in white and gray matter in GED patients using voxel-based morphometry. Compared to healthy controls, patients with GED had significantly less gray matter within the right middle frontal gyrus. Additionally, patients with GED had decreased fractional anisotropy and increased mean, axial, and radial diffusivities in the middle occipital gyrus, cuneus, and right superior occipital gyrus. The fractional anisotropy in these patients was shown to positively correlate with visual acuity (r = 0.456, *p* = 0.025). There was a negative correlation with the duration of disease (r = 0.609, *p* = 0.003). Microstructural and morphological abnormalities in areas corresponding to known functional deficits of cognition and vision were found in patients with GED. 

### 5.4. Diagnosis of Compressive Optic Neuropathy (CON)

Enlargement of EOMs, subsequent crowding at the apex of the orbit, and fat plane effacement around the optic nerve by the muscle are common radiological signs of CON [30,48,49,50,51,52]. The increased EOM volume often leads to exophthalmos, periorbital edema, venous congestion, and elevated intraocular pressure [53,54,55,56]. 

Stark et al. [28] found a correlation between the visual field with the worst mean deviation (MD) and EOM enlargement on CT in patients with CON. They demonstrated that as the EOM area increases on CT, the visual field MD worsens. Using multivariate linear regression, they identified that an increased superior rectus–levator muscle complex area can accurately predict visual field MD (*p* = 0.01) over the total EOM area (*p* = 0.25). They concluded that enlargement of the superior rectus–levator muscle complex may predict worsening CON.

MRI can also be used to evaluate CON. In a study by Dodds et al. [57] optic nerve diameters were measured and compared between three different patient populations: (1) patients who have GED and CON (GED^+^CON^+^), (2) patients who have GED but do not have CON (GED^+^CON^−^), and (3) control patients who do not have GED or CON (GED^−^CON^−^). Significant differences in optic nerve diameter were noted when comparing the GED^+^CON^+^ and GED^+^CON^−^ groups. No significant differences were found between the GED^+^CON^−^ and GED^−^CON^−^ groups. The intracranial pre-canalicular and intra-orbital apical segments of the optic nerves were the most commonly affected locations identified in their study. Additionally, there was optic nerve narrowing on MRI in some patients with GED with CON, whereas other patients in this same group presented with exophthalmos and typical EOM presentation. The authors proposed that optic nerve compression may be related to optic neuropathy caused by increased fat volume.

## 6. Radiological Differential Diagnosis

The radiologic differential diagnosis should include several diseases that commonly involve the EOMs. Ancillary findings are frequently required to arrive at the correct diagnosis.

Autoimmune diseases such as IgG4-related disease (IgG4-RD) (Figure 4) are characterized by diffuse lesions made up of lymphoplasmacytic infiltration containing storiform fibrosis and IgG4-positive plasma cells [58,59]. The lateral rectus has been reported to be the most commonly involved muscle, but this may not always be the case, and the tendinous insertions are often spared. IgG4-RD often presents with fat infiltration of the orbit, pre-septal soft-tissue swelling, and perineural involvement, which can include the maxillary branch of the trigeminal nerve (V_2_) [60]. Additional findings on imaging include infiltration of the pachymeninges, pituitary gland, lacrimal glands, salivary glands, and the thyroid gland [58].

Idiopathic orbital inflammation (IOI), previously named orbital pseudotumor, is caused by polymorphous lymphocyte infiltration with varying degrees of fibrosis, leading to painful inflammation of the orbit (Figure 5). In contradistinction to GED, IOI often involves tendinous portions of EOMs. Cases of IOI can also involve retro-bulbar fat, the optic nerve, the globe and optic nerve junction, the lacrimal glands, and the cavernous sinus [61,62].

Erdheim–Chester disease (ECD) is a rare multisystem histiocytic neoplasm. ECD is characterized by lipid-laden histiocyte tissue infiltration [63,64]. Imaging findings usually show bone involvement that can present as medullary sclerosis. Other areas of involvement may include the orbits and EOMs (Figure 6), meninges, brain parenchyma, paranasal sinuses, and the hypothalamic–pituitary axis. ECD also impacts the bony spinal column, mediastinum, retroperitoneum, and pulmonary interstitium [65,66,67]. 

Orbital lymphoma accounts for 2% of lymphomas, making it the most common primary orbital malignancy in adults [68]. The sites of involvement of orbital lymphoma include the EOMs and retrobulbar fat (Figure 7), eyelid, conjunctiva, and lacrimal gland. When lymphoma involves EOM, the presentation is usually painless and unilateral with associated tendon thickening and involvement of the tendinous insertion. MRI findings in patients with orbital lymphoma tend to be nonspecific with T1 and T2 homogeneous enhancement and an isointense signal intensity [69].

Orbital metastasis commonly involves the EOMs. Approximately 60% of EOM metastases cause EOM enlargement [70]. Additionally, about 58% of patients with orbital metastasis have already been diagnosed with a primary tumor. These primary tumors are most commonly gastrointestinal tumors, breast cancer, or cutaneous melanoma. While GED imaging findings tend to show diffuse infiltration, EOM metastasis findings show a nodular pattern (Figure 8) [71,72]. The number of EOMs affected can vary, and there may be a delay in time from the diagnosis of the primary disease. Other findings may include lesions in the bony orbit or the intracranial compartment [71]. 

EOM enlargement can also be caused by vascular conditions such as arteriovenous malformations, dural-venous shunts, venous angiomas, and carotid-cavernous sinus fistulas. Increased venous pressure and vascular distention can cause EOM enlargement, which may be diffuse or affect a single EOM [73]. Another differential consideration is that infection which spreads from the ethmoid sinus commonly involves the medial rectus muscle [74].

## 7. Treatment

GED can be self-limiting, and patients often recover spontaneously within 2–5 years [12]. When medical treatment is required, immunomodulators or radiotherapy can be used to limit immunologic damage during the early active phase [75]. In the post-inflammatory phase, surgery may be needed to improve patient comfort, preserve the orbit, and prevent vision loss from optic nerve compression or breakdown of the cornea [18]. Treatment tends to be relatively effective, and the recurrence rate of GED is less than 10% [76].

## Figures and Tables

**Figure 1 biomedicines-11-00312-f001:**
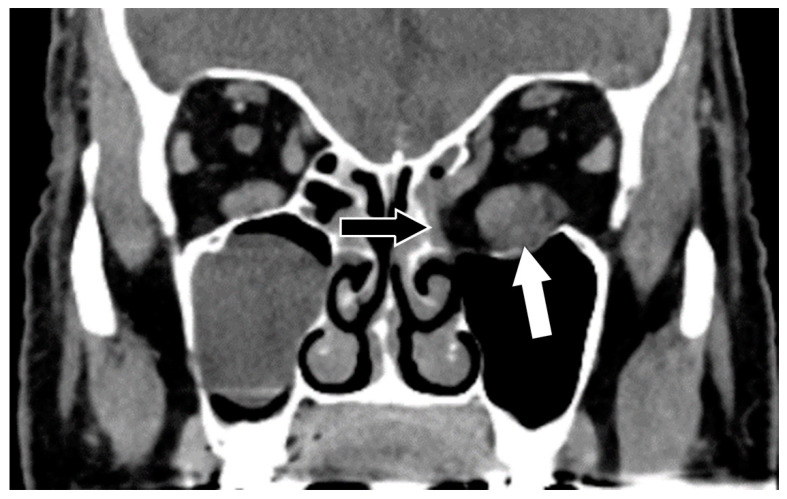
Graves’ eye disease. Coronal CT with contrast shows an enlarged left inferior rectus muscle (white arrow) and post-operative changes following decompressive surgery in the infero-medal left orbital wall and the medial aspect of the left orbital floor (black arrow).

**Figure 2 biomedicines-11-00312-f002:**
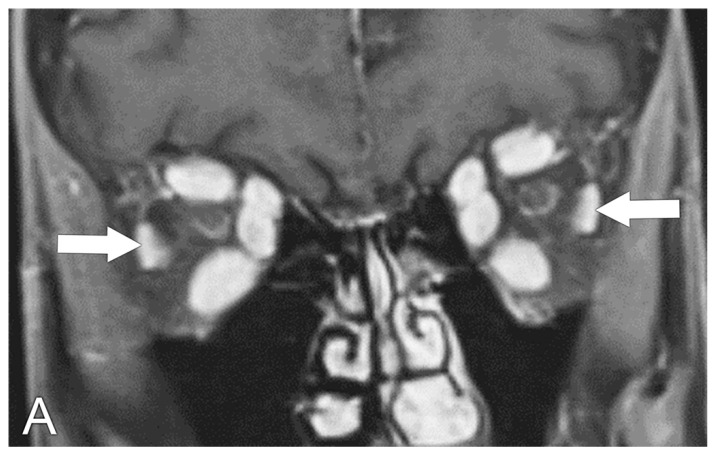
Graves’ eye disease, MRI. (**A**) Coronal T1 post-contrast MRI sequence shows enlargement of bilateral extra-ocular muscles with sparring of the lateral rectus muscles (arrows). (**B**) Coronal T2 MRI sequence shows hyperintense signal signifying edema in the enlarged extraocular muscles (arrow). (**C**) Coronal T1 post-contrast MRI sequence shows the normal appearance of the extra-ocular muscles.

**Figure 3 biomedicines-11-00312-f003:**
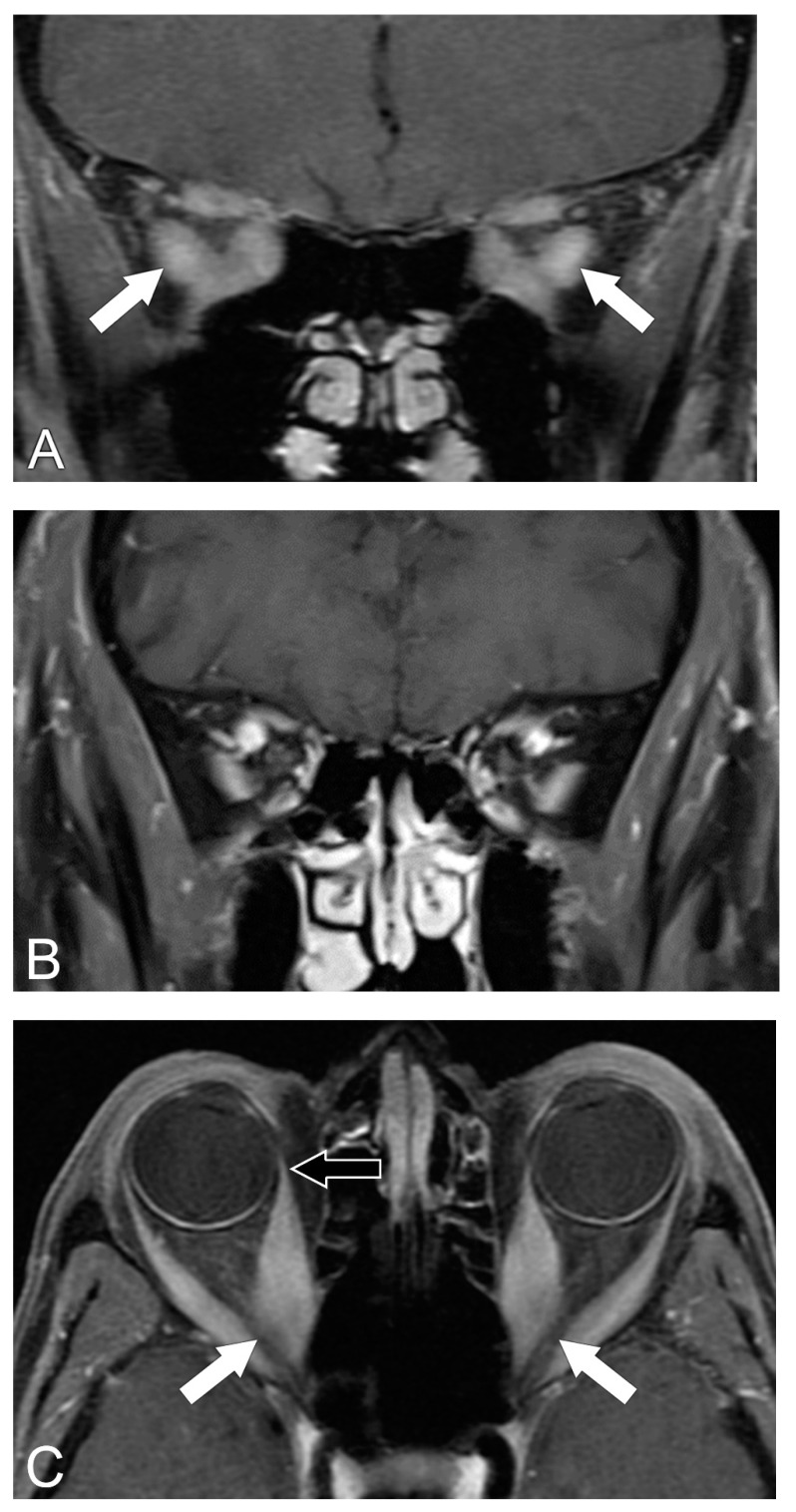
Compressive optic neuropathy. (**A**) Coronal T1 post-contrast MRI sequence shows enlargement of the extra-ocular muscles at the orbital apices (arrows). (**B**) Coronal T1 post-contrast MRI sequence shows the normal appearance of the extra-ocular muscles at the orbital apices. (**C**) Axial T1 post-contrast MRI sequence shows crowding of enlarged extra-ocular muscles with compression upon the optic nerves (white arrows). Note sparing of the tendinous insertions of the extraocular muscles (black arrow). (**D**) Axial T1 post-contrast MRI sequence shows a normal appearance of the extra-ocular muscles.

**Figure 4 biomedicines-11-00312-f004:**
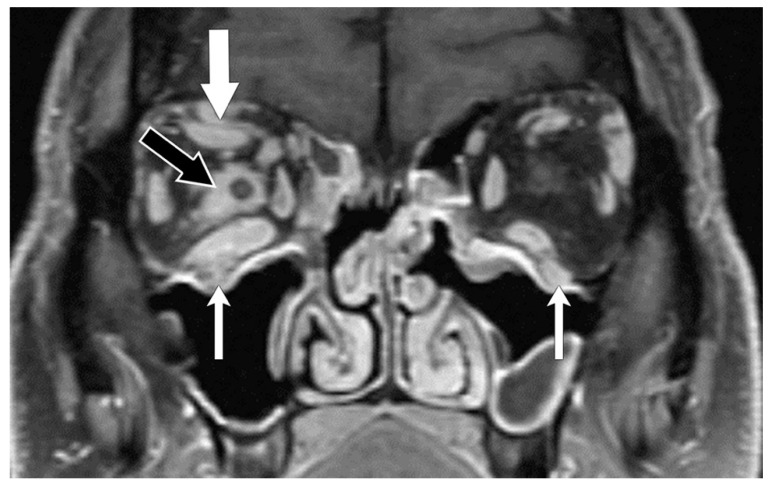
IgG4-related disease. Coronal T1 post-contrast MR sequence shows enlargement of the right superior rectus–levator complex (thick white arrow) and the inferior rectus muscle. Enhancing disease is also present around the optic nerve (black arrow) and seen involving the infra-orbital nerves bilaterally (V_2_) (thin white arrows).

**Figure 5 biomedicines-11-00312-f005:**
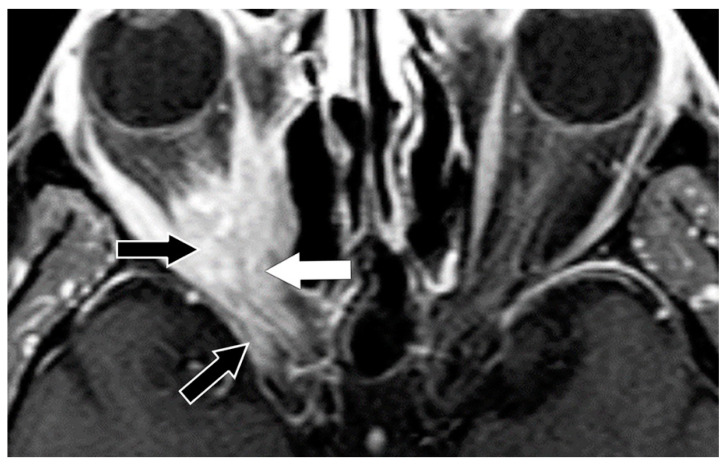
Idiopathic orbital inflammation (IOI). Axial T1 post-contrast MRI sequence shows enlargement of the right medial rectus muscle (white arrow). Additional enhancing disease is present in the right orbit with involvement of the tendinous insertions and extension through the superior orbital fissure towards the cavernous sinus (black arrows).

**Figure 6 biomedicines-11-00312-f006:**
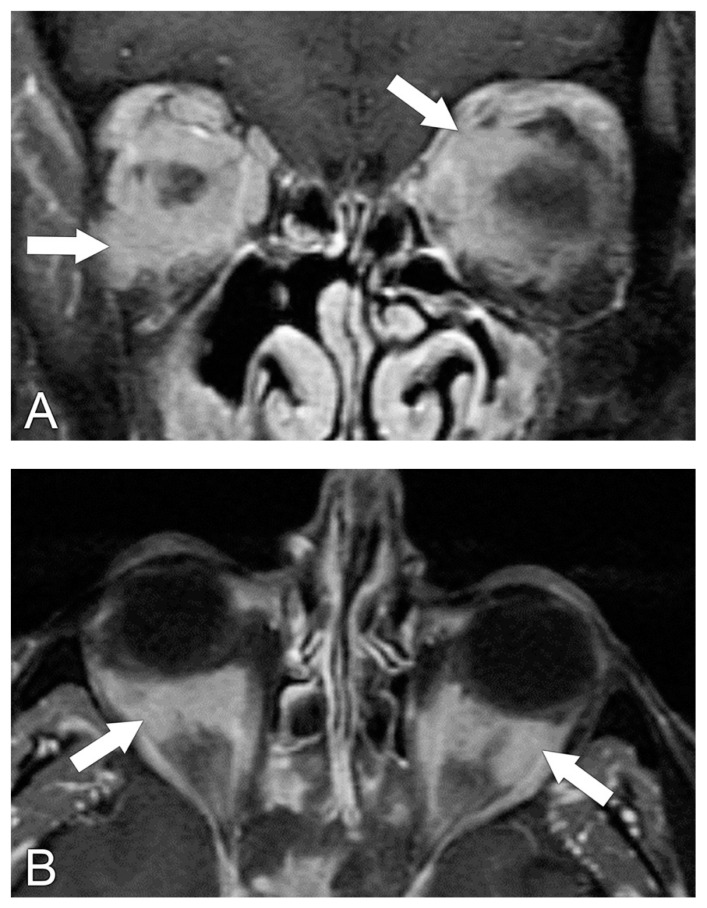
Erdheim–Chester disease. (**A**) Coronal T1 post-contrast MRI sequence shows extra-ocular muscle enlargement and disease in bilateral retrobulbar spaces (arrows). (**B**) Axial T1 post-contrast MRI sequence shows the disease in the bilateral intraconal spaces.

**Figure 7 biomedicines-11-00312-f007:**
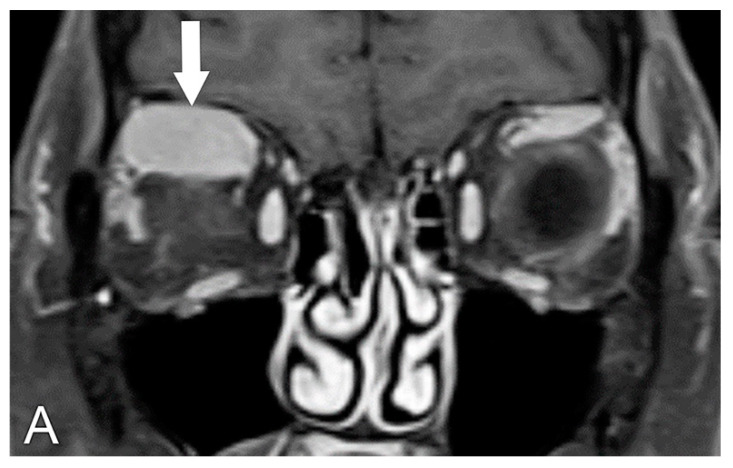
Lymphoma. (**A**) Coronal T1 post-contrast MRI sequence shows a homogeneously enhancing, well-circumscribed mass involving the right superior rectus–levator muscle complex (arrow). (**B**) Coronal PET image shows diffuse adenopathy throughout the neck and body.

**Figure 8 biomedicines-11-00312-f008:**
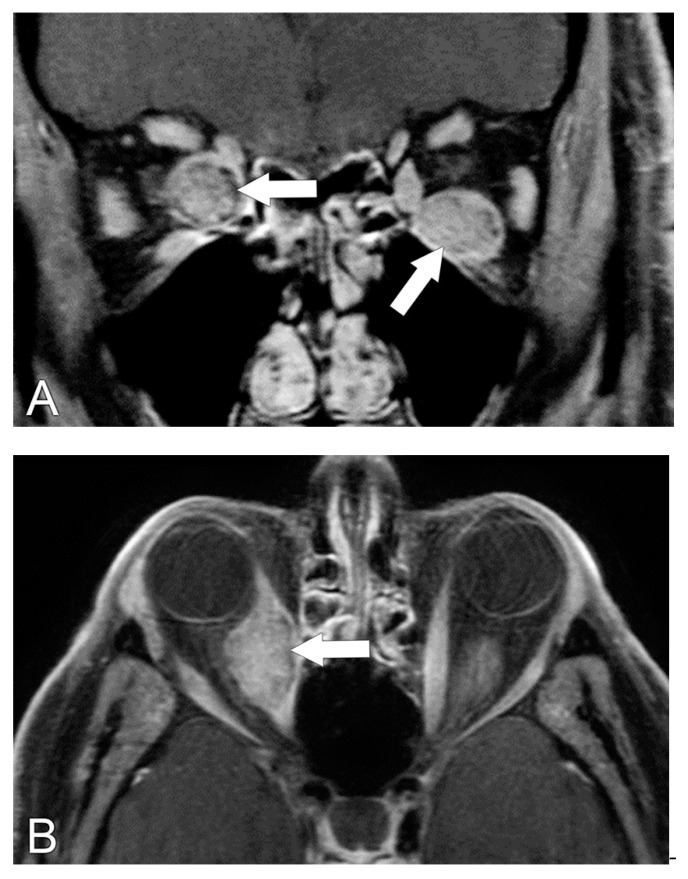
Gastrointestinal carcinoid tumor metastatic to the extra-ocular muscles. (**A**) Coronal T1 post-contrast MRI sequence shows heterogeneously enhancing metastases to the right medial and left inferior rectus muscles (arrows). (**B**) Axial T1 post-contrast sequence shows a nodular appearance of the right medial metastasis (arrow).

**Table 1 biomedicines-11-00312-t001:** Clinical activity score (CAS) *.

1—Orbital pain
2—Pain evoked by gaze
3—Eyelid swelling
4—Eyelid redness
5—Injected conjunctiva
6—Chemosis
7—Inflamed caruncle

* based on table by Mourits et al [32].

**Table 2 biomedicines-11-00312-t002:** NO SPECS classification of Graves’ Eye Disease *.

Classification	Signs and Symptoms (Grade)
0	Asymptomatic, no physical exam findings
I	Physical signs only
II	Involvement of soft tissue
	0: absent
	a: minimal
	b: moderate
	c: marked
III	Proptosis
	0: absent
	a: minimal: 3 to 4 mm above normal
	b: moderate: 5 to 7 mm above normal
	c: marked: 8+ mm above normal
IV	Extraocular muscles
	0: absent
	a: minimal: gaze limitations at extremes
	b: moderate: motion restriction, no fixation
	c: marked: globe fixation
V	Cornea
	0: absent
	a: minimal: stippling
	b: moderate: ulceration
	c: marked: necrosis, perforation, clouding
VI	Optic nerve compression/vision loss
	0: absent
	a: minimal: 20/20–20/60 vision; disc pallor
	b: moderate: 20/70–20/200 vision; disc pallor
	c: marked: <20/200 vision; blindness

* based on table by Werner [34].

**Table 3 biomedicines-11-00312-t003:** VISA (Vision, Inflammation, Strabismus, and Appearance) *.

• Inflamed caruncle	0: absent	1: present	
• Injected conjunctiva	0: absent	1: present	
• Eyelid redness	0: absent	1: present	
• Diurnal variation of symptoms	0: absent	1: present	
• Chemosis	0: absent		
	1: conjunctiva behind lid		
	2: conjunctiva beyond lid		
• Painful sensation behind globe			
At rest	0: absent	1: present	
With gaze	0: absent	1: present	
• Eyelid edema	0: absent		
	1: present without festoons (bags under eyes)		
	2: present with festoons		

* based on table by Barrio-Barrio et al [35].

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
