# Peer review of "Graves’ Eye Disease: Clinical and Radiological Diagnosis"

_biomedicines, 2023, doi:10.3390/biomedicines11020312_

Round 1

Reviewer 1 Report

This manuscript is somewhere between a case notes report to radiologists or an in-depth review of GED for a broader audience. This paper has many instances where jargon was used instead of a more descriptive explanation. At the risk of being more verbose, the readers could use explanation of the radiological terms. Some specifics are as follows.  

 Please ensure all acronyms are defined at first use.

In the introduction the first statement is weak, “Graves’ eye disease (GED) is an inflammatory condition of orbital soft tissues commonly 9 seen in patients with Graves’ disease.”

Try a statement like “Graves’ disease is an autoimmune disorder where the hyperthyroidism results in various systemic symptoms, with about 30% of patients presenting Graves’ eye disease (GED) is an inflammatory condition of orbital soft tissues.”

L165 Hounsfield unit – please define it is a radiodensity measurement with tissue density ranges defined. IE -1000HU is air vs 0HU as water and 1000HU for bone.

 MRI vs CT modalities could be further explained as both have uses in diagnostics and this is a case-use to highlight the pros and cons of each.

T1 and T2 weighted images for enhanced contrast are not defined for the reader. With the soft tissues the fat content is important to explain.

L302 “isointense signal intensity” with equivalent signal intensity, keeping definition and less jargon – define how contrast may be used to differentiate tissue.

With the images, a comparison vs healthy individuals could be useful. Colorized or segmented overlays may also be useful for the readers.

While the paper is well written, improvements can be made to be one of the best review/reference papers on GED diagnostics. Few other papers break down the clinical diagnostics procedures

Author Response

Dear Reviewer:

Thank you for your time and expertise in reviewing our manuscript. We feel that your suggestions have improved our paper. Please see our response to your comments and let us know if any further revisions are necessary.

 1. Please ensure all acronyms are defined at first use.

We double-checked that the acronyms are defined at first use.

2. In the introduction the first statement is weak, “Graves’ eye disease (GED) is an inflammatory condition of orbital soft tissues commonly 9 seen in patients with Graves’ disease.”

Try a statement like “Graves’ disease is an autoimmune disorder where the hyperthyroidism results in various systemic symptoms, with about 30% of patients presenting Graves’ eye disease (GED) is an inflammatory condition of orbital soft tissues.”

We changed the statement in lines 10-12 to "Graves’ disease is an autoimmune disorder in which hyperthyroidism results in various systematic symptoms, with about 30% of patients presenting with Graves’ eye disease (GED)."

3. L165 Hounsfield unit – please define it is a radiodensity measurement with tissue density ranges defined. IE -1000HU is air vs 0HU as water and 1000HU for bone.

We defined and gave ranges for Hounsfield units with additional text in lines 166-169.

 3. MRI vs CT modalities could be further explained as both have uses in diagnostics and this is a case-use to highlight the pros and cons of each.

We added this description with additional text in lines 164-165 and 178.

4. T1 and T2 weighted images for enhanced contrast are not defined for the reader. With the soft tissues the fat content is important to explain.

We addressed this topic with additional text in lines 179-186.

5. L302 “isointense signal intensity” with equivalent signal intensity, keeping definition and less jargon – define how contrast may be used to differentiate tissue.

We addressed this topic in line 183.

6. With the images, a comparison vs healthy individuals could be useful. Colorized or segmented overlays may also be useful for the readers.

We added normal images for comparison to Figures 2C (text lines202-203) and Figures 3B and 3D (text lines 212-213 and 216-217). Figure 3B from the first submission was moved to become Figure 3C.

Reviewer 2 Report

hello

very nice paper

I have nothing to add, it expresses the exact amount of valuable data

great work,

with regards

Author Response

Dear Reviewer:

Thank you for your time and expertise in reviewing our manuscript! 

Sincerely

Reviewer 3 Report

No major concerns or issues; well written review. Minor English revision would be beneficial.  

Author Response

Dear Reviewer:

Thank you for your time and expertise in reviewing our manuscript! 

We have checked the English in the manuscript.

Sincerely